# PrefPoE: Advantage-Guided Preference Fusion for Learning Where to Explore

## Abstract

Exploration in reinforcement learning remains a critical challenge, as naive entropy maximization often results in high variance and inefficient policy updates. We introduce **PrefPoE**, a novel *Preference-Product-of-Experts* framework that performs intelligent, advantage-guided exploration via the first principled application of product-of-experts (PoE) fusion for single-task exploration-exploitation balancing. By training a preference network to concentrate probability mass on high-advantage actions and fusing it with the main policy through PoE, PrefPoE creates a **soft trust region** that stabilizes policy updates while maintaining targeted exploration. Across diverse control tasks spanning both continuous and discrete action spaces, PrefPoE demonstrates consistent improvements: +321% on HalfCheetah-v4 ($1276 \rightarrow 5375$), +69% on Ant-v4, +276% on LunarLander-v2, with consistently enhanced training stability and sample efficiency. Unlike standard PPO, which suffers from entropy collapse, PrefPoE sustains adaptive exploration through its unique dynamics, thereby preventing premature convergence and enabling superior performance. Our results establish that learning *where to explore* through advantage-guided preferences is as crucial as learning how to act, offering a general framework for enhancing policy gradient methods across the full spectrum of reinforcement learning domains. Code and pretrained models are available in supplementary materials.

## 1 Introduction

Reinforcement learning (RL) presents a fundamental exploration-exploitation trade-off. While off-policy methods such as Soft Actor-Critic (SAC) (Haarnoja et al., 2018) benefit from replay buffers and entropy-based exploration, efficient exploration remains a particular challenge for on-policy algorithms. In particular, algorithms like Proximal Policy Optimization (PPO) (Schulman et al., 2017) must balance exploration and exploitation within each policy update (Tang et al., 2021; Lyu et al., 2022; Seo et al., 2025). While policy gradient methods like PPO and Trust Region Policy Optimization (TRPO) (Schulman et al., 2015a) have achieved remarkable success due to their stability and ease of implementation, they remain fundamentally limited by inefficient exploration strategies that treat all regions of the action space with equal importance.

Current exploration strategies in RL often rely on entropy regularization (Haarnoja et al., 2018) or uniform action noise (Lillicrap et al., 2015), both of which perform broad, undirected "spray-and-pray" sampling that becomes inefficient as dimensionality grows. In RL tasks, such random search struggles to find high-value actions, leading to poor sample efficiency and high-variance updates. Moreover, unlike SAC, which integrates entropy into the Bellman backup (Sutton & Barto, 1998) for *first-order* optimization, PPO can only apply a small *second-order* entropy bonus that quickly becomes negligible as rewards dominate, causing premature convergence.

The key insight driving our work is that **exploration should be guided by advantage estimates rather than being uniformly distributed**. If we can identify regions of the action space that consistently yield higher rewards, why not concentrate our exploration efforts there? This principle suggests moving from "exploration everywhere" to "exploration where it matters most"—from quantity to quality in action sampling. Our empirical results validate this view: focused exploration with lower entropy can in fact outperform random exploration with higher entropy, challenging the conventional wisdom that more entropy necessarily leads to better exploration.

We implement this principle through a novel RL training framework **PrefPoE** (*Preference-Product-of-Experts*), which has a two-branch architecture. It employs a shared backbone architecture with two policy heads: a *main policy head* and a *preference head* that learns to concentrate probability mass in high-advantage action regions. These are fused using a mathematically principled Product-of-Experts (PoE) mechanism (Hinton, 2002). This PoE fusion sharpens the resulting action distribution by emphasizing regions favored by both policy heads, balancing exploitation and exploration. It results in an intelligent exploration mechanism that maintains sampling diversity while focusing exploration where it matters most.

The key technical innovation lies in our advantage-guided learning objective for the preference network. Rather than mimicking successful actions directly, the preference network learns to assign higher probabilities to actions with higher advantage estimates, naturally implementing a Boltzmann distribution (Wang et al., 2020) over advantage values. Combined with the main policy through PoE fusion, this produces a **soft trust region**: unlike the hard KL constraints used in TRPO and PPO, this region emerges organically from the variance-reducing property of PoE, stabilizing policy updates while preserving exploration diversity. In contrast to uniform exploration strategies (Schulman et al., 2017; 2015a) that treat all actions equally, PrefPoE leverages advantage-weighted preferences to concentrate exploration on promising regions. As a modular extension, it complements existing policy gradient methods by unifying value-based guidance with policy-based execution to enable more focused and efficient exploration.

Our experimental evaluation demonstrates the effectiveness and generality of PrefPoE across diverse control challenges. Beyond achieving consistent performance improvements on standard benchmarks like HalfCheetah-v4 and Ant-v4 (Todorov et al., 2012), PrefPoE exhibits two critical advantages: enhanced training stability through adaptive exploration dynamics, and seamless applicability to both continuous and discrete action spaces without architectural modifications.

Our work makes the following key contributions:

- **Conceptual Innovation:** We introduce the principle of advantage-guided exploration, demonstrating that focused, lower-entropy exploration can outperform uniform, high-entropy sampling—challenging conventional RL wisdom.
- **Technical Framework:** We present PrefPoE, the first systematic application of Product-of-Experts fusion for single-task exploration-exploitation balance, implementable as a plug-and-play enhancement for any policy gradient method.
- **Theoretical Analysis:** We show that the preference learning converges to Boltzmann distributions over advantages and prove that PoE fusion creates soft trust regions through variance reduction.
- **Broad Applicability:** We achieve consistent improvements across continuous (+321% on HalfCheetah) and discrete (+276% on LunarLander) domains, demonstrating both generality and robustness to entropy collapse.

The rest of the paper is organized as follows: Section 2 reviews related work on exploration strategies, Section 3 details our PrefPoE framework, Section 4 provides theoretical analysis, Section 5 presents experimental results, and Section 6 provides discussion and conclusion.

## 2 RELATED WORK

**1) Exploration in Continuous Control.** Exploration remains one of the most fundamental challenges in RL (Nair et al., 2018), particularly in continuous action spaces where naive random sampling (e.g., uniform action noise) and undirected entropy maximization both become increasingly inefficient as dimensionality grows. Traditional approaches have largely fallen into two categories: *parameter space exploration*, and *action space exploration*.

Parameter space methods, such as evolutionary strategies (Salimans et al., 2017) and parameter noise (Plappert et al., 2017), add noise to policy parameters. While effective in some domains, they lack the fine-grained control over action selection our advantage-guided approach provides.

Action space methods, including epsilon-greedy variants for continuous control (Dulac-Arnold et al., 2019) and Ornstein-Uhlenbeck noise (Lillicrap et al., 2015), perturb actions directly but typ-

ically employ uniform noise distributions that ignore the underlying value landscape. More sophisticated exploration strategies utilize entropy-based regularization. SAC (Haarnoja et al., 2018) incorporates entropy maximization directly into the Bellman equation (Sutton & Barto, 1998), enabling *first-order* entropy optimization and maintaining exploration pressure throughout training. In contrast, PPO (Schulman et al., 2017) adds an entropy bonus $\beta H(\pi)$ only as a small *second-order* regularization term ($\beta \ll 1$). As rewards dominate, this bonus quickly becomes negligible, often leading to exploration collapse. Both methods also treat all actions within the policy's support equally, potentially wasting effort on low-value regions. In contrast, our approach performs targeted action space exploration by learning a preference distribution that concentrates exploration in high-advantage regions. The preference network's entropy regularization ensures exploration diversity, while the PoE fusion sharpens the final distribution toward mutually agreed high-value actions.

**2) Policy Gradient Enhancements.** The policy gradient family of algorithms have seen extensive work on improving sample efficiency and training stability. Trust region methods like TRPO (Schulman et al., 2015a) and PPO (Schulman et al., 2017) constrain policy updates to maintain stability but use uniform constraints that do not differentiate between high- and low-quality actions. Recent research explores adaptive or learned constraints. Meta-learning approaches (Finn et al., 2017) learn to adapt quickly to new tasks but require multiple environments. Natural policy gradients (Kakade, 2001; Liu et al., 2023) adjust update directions more effectively but still rely on uniform exploration. Our work complements these advances by providing a principled way to bias exploration toward promising regions while preserving the stability benefits of existing methods.

**3) Product-of-Experts and Mixture Models.** PoE (Hinton, 2002) is a well-established framework for combining multiple probability distributions, originally used in unsupervised learning. While the PoE method has been extensively studied in unsupervised learning, its application to policy fusion in RL remains largely unexplored. In RL, mixture models and expert combinations have mainly focused on multi-task (Ghosh et al., 2018) or hierarchical policies (Hausman et al., 2018). However, prior work has not explored PoE for combining exploration and exploitation within a single policy. Our work represents the first systematic investigation of using PoE to improve exploration in RL. Our approach differs fundamentally by using PoE to fuse a learned preference distribution with the main policy, resulting in a hybrid that emphasizes areas of agreement. Unlike simple mixture models—whether based on averaging or attention-weighted combinations of experts (Cheng et al., 2023; Obando-Ceron et al., 2024)—PoE naturally sharpens the final distribution around high-confidence regions. This variance-reducing property effectively creates *soft trust regions*, making PoE particularly suitable for balancing exploration and exploitation without requiring explicit constraints.

**4) Advantage-Based Methods.** Advantage estimation has long been central to policy gradient methods, starting with early Actor-Critic algorithms (Konda & Tsitsiklis, 1999). To improve bias-variance trade-offs, Generalized Advantage Estimation (GAE) (Schulman et al., 2015b) was introduced. Recent work further refines advantage usage through techniques such as normalization (Engstrom et al., 2020) and adaptive scaling (Song et al., 2019), aiming to stabilize learning and improve sample efficiency. In contrast to prior work (Wagenmaker et al., 2023), we propose a novel use of advantage information: rather than using advantages solely for gradient updates, we leverage them to learn where exploration should concentrate. This creates a feedback loop—better value estimates enable efficient exploration, which in turn accelerates learning and enhances policy robustness.

## 3 PROBLEM FORMULATION AND METHODOLOGY

### 3.1 PROBLEM FORMULATION AND SYSTEM STRUCTURE

We consider a standard continuous control Markov Decision Process (MDP) (Bellman, 1957) defined by the tuple $(\mathcal{S}, \mathcal{A}, P, R, \gamma)$, where $\mathcal{S}$ is the state space, $\mathcal{A} \subset \mathbb{R}^d$ is the continuous action space, $P : \mathcal{S} \times \mathcal{A} \to \Delta(\mathcal{S})$ denotes the state transition kernel, where $\Delta(\mathcal{S})$ is the probability simplex over $\mathcal{S}$, $R : \mathcal{S} \times \mathcal{A} \to \mathbb{R}$ is the reward function, and $\gamma \in [0, 1)$ is the discount factor. The design goal is to find a control policy $\pi$ that maximizes the expected return $J(\pi) = \mathbb{E}_\pi[\sum_{t=0}^\infty \gamma^t R(s_t, a_t)]$.

In typical policy gradient methods, the policy $\pi$ is parameterized as a multivariate Gaussian: $\pi_\theta(a|s) = \mathcal{N}(\mu_\theta(s), \Sigma_\theta(s))$, where $\mu_\theta(s) \in \mathbb{R}^d$ and $\Sigma_\theta(s) \in \mathbb{R}^{d \times d}$ denote the mean and covariance output by neural networks, and $\theta$ denotes the neural networks parameters. Exploration

is usually encouraged by adjusting $\Sigma_\theta(s)$ or adding an entropy bonus, but such strategies treat all action regions uniformly, leading to inefficient exploration in RL tasks.

We present PrefPoE (see Figure 1), a plug-and-play framework that enhances policy gradient methods through advantage-guided exploration. The architecture employs a shared backbone with two specialized policy heads: a main policy head for return maximization and a preference head that learns to concentrate probability mass on high-advantage actions. The shared representation ensures consistent state encoding while enabling specialization: the main policy focuses on return maximization, whereas the preference network learns to prioritize where exploration should be concentrated. These policies are fused via a principled PoE mechanism to produce the final action distribution. During training, the advantage estimates from environment interactions guide the preference network learning, creating a feedback loop that adaptively sharpens exploration in promising regions while maintaining stability. Details of PrefPoE are provided below.

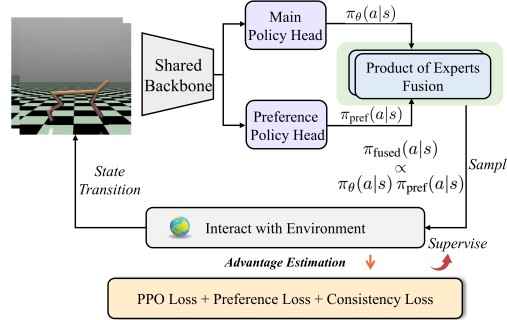

Figure 1: Architecture of PrefPoE. A shared backbone (a common encoder $f_{\text{enc}}(s)$) feeds into main and preference policy heads, which are fused via PoE to generate actions. The preference head is trained with advantage guidance to focus exploration on high-value regions.

### 3.2 ADVANTAGE-GUIDED PREFERENCE LEARNING

Our key insight is that exploration should be advantage-guided rather than uniform. To this end, we introduce a preference network $\pi_{\text{pref}}(a|s)$ that learns a policy density that concentrates probability mass on high-advantage actions, given by

$$\pi_{\text{pref}}(a|s) = \mathcal{N}(\mu_{\text{pref}}(s), \Sigma_{\text{pref}}(s)), \ \mu_{\text{pref}}(s) = f_\mu^{\text{pref}}(f_{\text{enc}}(s)), \ \sigma_{\text{pref}}(s) = \exp\left(f_\sigma^{\text{pref}}(f_{\text{enc}}(s))\right), \quad (1)$$

where $f_\mu^{\text{pref}}$ and $f_\sigma^{\text{pref}}$ are Multi-Layer Perceptron (MLP) heads that output the mean and log-standard deviation; $\sigma_{\text{pref}}(s)$ is obtained by exponentiating the log-std output, and $\Sigma_{\text{pref}}(s) = \text{diag}(\sigma_{\text{pref}}(s)^2)$ is the resulting diagonal covariance matrix. $f_{\text{enc}}(s)$ is the common encoder shared by the main policy and preference networks.

We propose an *advantage-guided preference learning strategy*, in which the preference network is trained with an objective function that assigns greater probability mass to actions with higher advantages, defined as follows:

$$\mathcal{L}_{\text{pref}}(\phi) = -\beta_1 \, \mathbb{E}_{(s,a)}\left[A_{\text{norm}}(s,a) \cdot \log \pi_{\text{pref}}(a|s)\right] - \alpha \, \mathcal{H}(\pi_{\text{pref}}), \quad (2)$$

where $\phi$ denotes the parameters of the preference network. $A_{\text{norm}}(s,a)$ is the batch-normalized advantage estimate, e.g., from GAE (Schulman et al., 2015b). $\mathcal{H}(\pi_{\text{pref}})$ is the entropy and for a diagonal Gaussian, it has the closed-form expression: $\mathcal{H}(\pi_{\text{pref}}) = \frac{1}{2} \sum_i \log(2\pi e \, \sigma_{\text{pref},i}^2)$. The hyperparameters $\beta_1$ and $\alpha$ control the strength of advantage guidance and entropy regularization, respectively.

Minimizing $\mathcal{L}_{\text{pref}}$ in (2) with respect to $\pi_{\text{pref}}$ yields a trade-off between advantage-weighted likelihood and entropy. This convex optimization admits a closed-form solution in the exponential family (see Theorem 1 and proof in Appendix E.1). The optimal preference $\pi_{\text{pref}}^*(a|s)$ density satisfies

$$\pi_{\text{pref}}^*(a|s) \propto \exp\left(\beta_1 A_{\text{norm}}(s,a)/\alpha\right). \quad (3)$$

This solution yields a Boltzmann distribution over advantages, concentrating exploration on promising actions while preserving diversity.

### 3.3 PRODUCT-OF-EXPERTS FUSION

We employ PoE fusion to merge the main policy $\pi_\theta(a|s) = \mathcal{N}(\mu_\theta(s), \Sigma_\theta(s))$, where $\mu_\theta(s)$ and $\Sigma_\theta(s)$ denote the mean and covariance of the main policy, with the learned preference $\pi_{\text{pref}}(a|s) =$

$\mathcal{N}(\mu_{\text{pref}}(s), \Sigma_{\text{pref}}(s))$ given in (2). For multivariate Gaussians, the unnormalized fused policy obtained via PoE is defined as

$$\pi_{\text{fused}}(a|s) \propto \pi_\theta(a|s) \cdot \pi_{\text{pref}}(a|s). \tag{4}$$

The fused distribution is then obtained by

$$\Sigma_{\text{fused}}^{-1} = \Sigma_\theta^{-1} + \lambda_{\text{pref}}\Sigma_{\text{pref}}^{-1}, \quad \mu_{\text{fused}} = \Sigma_{\text{fused}}(\Sigma_\theta^{-1}\mu_\theta + \lambda_{\text{pref}}\Sigma_{\text{pref}}^{-1}\mu_{\text{pref}}), \tag{5}$$

where $\lambda_{\text{pref}} \in (0, 1)$ controls the influence of the preference network. This formulation emphasizes regions where both experts agree, yielding a sharper and more confident distribution. By construction, the fused covariance satisfies $\text{tr}(\Sigma_{\text{fused}}) \leq \min(\text{tr}(\Sigma_\theta), \text{tr}(\Sigma_{\text{pref}}))$, acting as a **soft trust region**.

### 3.4 TRAINING FRAMEWORK AND EXTENSIONS

**1) Integration with PPO.** The proposed PrefPoE integrates seamlessly with PPO. Instead of sampling actions from the main policy $\pi_\theta(a|s)$, we sample from the fused distribution $\pi_{\text{fused}}(a|s)$ in (4). The total loss $\mathcal{L}_{\text{total}}$ is defined as

$$\mathcal{L}_{\text{total}} = \mathcal{L}_{\text{PPO}} + w_{\text{pref}}\mathcal{L}_{\text{pref}} + w_{\text{cons}}\mathcal{L}_{\text{cons}}, \tag{6}$$

where $\mathcal{L}_{\text{PPO}}$ is the standard PPO clipped surrogate objective, $\mathcal{L}_{\text{pref}}$ is the advantage-guided preference loss in (2), and $\mathcal{L}_{\text{cons}} = D_{\text{KL}}(\pi_{\text{fused}}(a|s)\|\pi_{\text{pref}}(a|s))$ aligns the fused policy with the preference network to maintain coherent exploration, whose strength are controlled by the coefficient $w_. \in (0, 1)$.

**2) Extension to discrete spaces.** The proposed PrefPoE applies seamlessly to discrete action spaces. Specifically, we replace Gaussian distributions with categorical distributions, and perform PoE fusion directly in probability space rather than precision space, as follows:

$$\pi_{\text{fused}}(a|s) = \frac{\pi_\theta(a|s) \cdot \pi_{\text{pref}}(a|s)^{\lambda_{\text{pref}}}}{\sum_{a' \in \mathcal{A}} \pi_\theta(a'|s) \cdot \pi_{\text{pref}}(a'|s)^{\lambda_{\text{pref}}}}. \tag{7}$$

The advantage-guided learning objective and Boltzmann-form preference policy remain unchanged, confirming that PrefPoE is structurally domain-agnostic. We provide theoretical analysis of this extension in Appendix D and experimental evaluation in Section 5.

## 4 THEORETICAL ANALYSIS

In this section, we provide theoretical foundations for PrefPoE, analyzing three key properties: the convergence of advantage-guided preference learning, the numerical stability of PoE fusion, and the gradient efficiency of advantage-directed exploration.

**1) Convergence of Advantage-guided Preference Learning.** Theorem 1 shows that our preference learning has a well-defined optimal solution, with detailed proof in Appendix E.1.

**Theorem 1.** *Assume that the advantage function $A_{norm}(s, a)$ is bounded by $|A_{norm}(s, a)| \leq A_{\max}$ for all $(s, a)$, and let $\phi$ denote the parameters of the preference network $\pi_{pref}(a|s)$. Then the loss function (2) admits a unique global minimum given by the Boltzmann distribution:*

$$\pi_{pref}^*(a|s) = \frac{\exp(\beta_1 A_{norm}(s, a)/\alpha)}{Z(s)}, \tag{8}$$

*where $Z(s) = \int \exp(\beta_1 A_{norm}(s, a)/\alpha)\mathrm{d}a$ is the partition function.*

Theorem 1 confirms that the preference network naturally concentrates probability mass on high-advantage actions, recovering a Boltzmann policy that balances exploitation and exploration through the temperature ratio $\alpha/\beta_1$.

**2) Numerical Stability of PoE Fusion.** Theorem 2 shows the stability properties of our Product-of-Experts fusion mechanism.

**Theorem 2.** *Let $\pi_\theta(a|s) = \mathcal{N}(\mu_\theta, \Sigma_\theta)$ and $\pi_{pref}(a|s) = \mathcal{N}(\mu_{pref}, \Sigma_{pref})$ be two Gaussian distributions. Then the PoE fusion satisfies the following properties:*

*1) Well-definedness: $\Sigma_{fused} = (\Sigma_\theta^{-1} + \lambda_{pref}\Sigma_{pref}^{-1})^{-1}$ is positive definite.*

*2) Variance reduction: $tr(\Sigma_{fused}) \leq \min(tr(\Sigma_\theta), \lambda_{pref}^{-1} tr(\Sigma_{pref}))$.*

*Proof.* Since $\Sigma_\theta \succ 0$ and $\Sigma_{\text{pref}} \succ 0$, their inverses are positive definite. With $\lambda_{\text{pref}} > 0$, the sum $\Sigma_\theta^{-1} + \lambda_{\text{pref}}\Sigma_{\text{pref}}^{-1} \succ 0$ is positive definite, ensuring that $\Sigma_{\text{fused}}$ exists and is positive definite.

Let $P = \Sigma_\theta^{-1}$ and $Q = \lambda_{\text{pref}}\Sigma_{\text{pref}}^{-1}$, with positive definite matrices $P$ and $Q$. According to Horn & Johnson (2012), $(P + Q)^{-1} \preceq P^{-1}$ and $(P + Q)^{-1} \preceq Q^{-1}$ holds. Then we can derive

$$\Sigma_{\text{fused}} = (P + Q)^{-1} \preceq P^{-1} = \Sigma_\theta, \quad \Sigma_{\text{fused}} \preceq Q^{-1} = \lambda_{\text{pref}}^{-1}\Sigma_{\text{pref}}. \tag{9}$$

According to Horn & Johnson (2012), the trace is monotonic with respect to the positive semidefinite ordering, i.e., $A \preceq B$ implies $\text{tr}(A) \leq \text{tr}(B)$. We can obtain from (9) that

$$\text{tr}(\Sigma_{\text{fused}}) \leq \min(\text{tr}(\Sigma_\theta), \lambda_{\text{pref}}^{-1}\,\text{tr}(\Sigma_{\text{pref}})). \tag{10}$$

Therefore, PoE fusion reduces the total variance compared to either individual distribution. □

While Theorem 2 provides theoretical guarantees, we further stabilize PoE fusion through two key regularization mechanisms. First, the consistency term $\mathcal{L}_{\text{cons}} = D_{\text{KL}}(\pi_{\text{fused}} \| \pi_{\text{pref}})$ in (6) anchors the fused policy toward the preference distribution, preventing degenerate solutions and ensuring numerical stability. Second, entropy regularization $\mathcal{H}(\pi_{\text{pref}})$ in the preference loss (2) prevents $\pi_{\text{pref}}$ from collapsing, maintaining the positive definiteness required by Theorem 2. Together, these mechanisms ensure that PoE fusion remains numerically stable and variance-reducing in training.

**3) Exploration Benefits via Advantage-Guided Sampling.** While a complete theoretical analysis of exploration efficiency remains challenging, we provide theoretical insights supported by empirical evidence. The preference distribution $\pi_{\text{pref}}(a|s)$ follows a Boltzmann form over $A_{\text{norm}}(s, a)$, enabling the fused policy $\pi_{\text{fused}}$ to assign greater probability mass to high-advantage actions—especially beneficial during early training when exploration is critical. According to Theorem 1, our preference entropy implements *guided entropy*: $\pi_{\text{pref}}^*(a|s) \propto \exp(\beta_1 A_{\text{norm}}(s, a)/\alpha)$, which concentrates exploration along advantage gradients rather than uniform sampling. This explains why PrefPoE achieves superior performance despite lower final entropy than PPO—**structured low-entropy exploration can be more valuable than unstructured high-entropy exploration.**

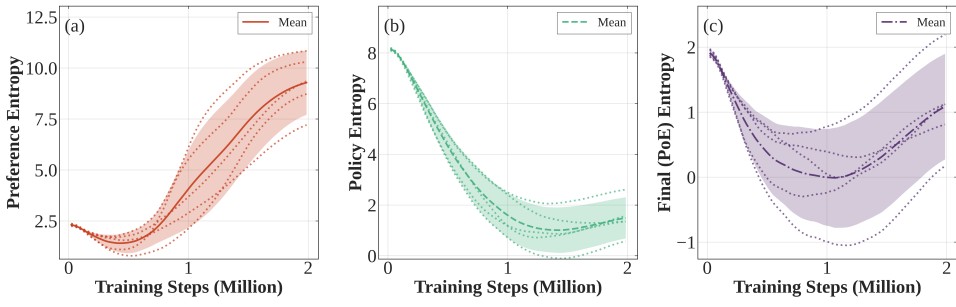

Figure 2: Entropy dynamics in HalfCheetah-v4 demonstrating PrefPoE's resistance to entropy collapse. (a) Preference entropy shows initial focusing and later broadening. (b) PPO entropy decays monotonically, converging near 1.0. (c) PoE entropy shows adaptive U-shaped dynamics, recovering and stabilizing around 1.0 (1–2M), thereby preventing the collapse observed in vanilla PPO.

Figure 2 illustrates the adaptive nature of our exploration strategy. The preference entropy (Figure 2(a)) decreases early on as the network concentrates on high-advantage actions, then increases moderately to maintain diversity. In contrast, policy entropy (Figure 2(b)) follows the typical PPO decay. The fused PoE entropy (Figure 2(c)) recovers to a stable level around 1.0, demonstrating controlled exploration. **Importantly, while the final entropy is lower than vanilla PPO (1.0 vs. 1.5), this reflects more efficient exploration rather than exploration failure.** The guided sampling amplifies the effective learning signal: $\mathbb{E}_{a\sim\pi_{\text{fused}}}[A(s, a)] > \mathbb{E}_{a\sim\pi_{\text{uniform}}}[A(s, a)]$, leading to more informative gradients and faster convergence. This validates our core hypothesis: focused exploration in high-value regions can be more effective than uniform high-entropy exploration. Analysis in Figure A.2 (Appendix) shows that the policy and PoE means remain closely aligned, with only 4.2% deviation, confirming stability without excessive conservatism.

**4) Implications for Training Dynamics.** Theoretical results above establish that PrefPoE offers key guarantees for stable and efficient learning: 1) advantage-guided preference learning converges to a well-defined Boltzmann distribution, preventing exploration collapse; 2) PoE fusion induces implicit regularization via variance reduction, serving as a soft trust-region mechanism; and 3) advantage-directed sampling concentrates updates on informative state-action pairs. Together, these properties explain why PrefPoE improves both sample efficiency and stability. Unlike fixed exploration schedules that decay uniformly, PrefPoE dynamically adjusts exploration intensity via advantage estimates—maintaining diversity in promising regions while suppressing noise in low-value areas.

## 5 EXPERIMENTS

We evaluate PrefPoE across continuous and discrete control tasks, demonstrating consistent performance gains and improved stability through comparisons with PPO baselines and ablations.

**Environments.** We evaluate on six diverse tasks: three continuous control environments (HalfCheetah-v4, LunarLanderContinuous-v2, Ant-v4) spanning locomotion, precision landing, and quadrupedal coordination; and three discrete control tasks (CartPole-v1, LunarLander-v2, FrozenLake-v1) covering classic control, precision landing, and sparse reward navigation.

**Baselines.** We compare against: 1) *Vanilla PPO* with standard entropy regularization (Huang et al., 2022), 2) *PPO + Adaptive Clipping* with linearly annealed clip ratios, and 3) *PPO + Linear Fusion* that averages policy outputs instead of using PoE, serving as an ablation for our fusion mechanism.

**Implementation.** All methods share the identical network architecture (2-layer MLP, 64 units) and training configurations from CleanRL (Huang et al., 2022). We report mean episodic returns $\pm$ standard deviation over 5 seeds {0, 10, 42, 77, 123}. See Appendix A.2 for hyperparameters and Appendix C for evaluation protocols.

### 5.1 VALIDATION ON CONTINUOUS ACTION SPACES

Table 1: Final performance across continuous control environments (mean $\pm$ std over 5 seeds).

| Method | HalfCheetah-v4 | LunarLander-v2 | Ant-v4 |
|---|---|---|---|
| PPO Baseline | 1276±67 | 148±70 | 2668±1367 |
| PPO + Adaptive Clipping | 1454±771 | 122±108 | 3201±574 |
| PPO + Linear Fusion | 1671±878 | 95±16 | 785±461 |
| PrefPoE (Ours) | **5375±581** | **217±59** | **4499±602** |
| Average Improvement | **+321%** | **+47%** | **+69%** |

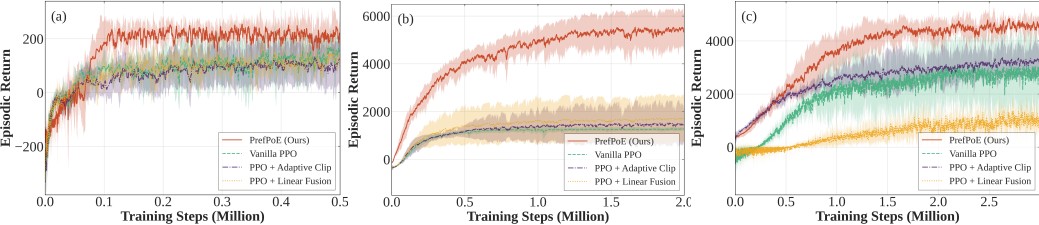

Figure 3: Learning curves on (a) LunarLanderContinuous-v2, (b) HalfCheetah-v4, and (c) Ant-v4.

Table 1 summarizes the main results across all evaluated environments. PrefPoE consistently outperforms all baselines, achieving substantial gains of +321% on HalfCheetah-v4, +47% on LunarLanderContinuous-v2, and +69% on Ant-v4. Beyond reward improvements, PrefPoE also enhances training stability: on HalfCheetah-v4, it maintains low variability (CV: 10.8%) despite high returns, and on Ant-v4, it significantly reduces the coefficient of variation from 51.2% (baseline PPO) to 13.4%. In comparison, PPO with adaptive clipping yields modest average improvements (+14%), while linear fusion performs inconsistently, suffering from instability in Ant and LunarLander (-12% overall). PrefPoE achieves the largest and most stable gains (+146% average). These

results highlight the benefit of aligning exploration with advantage structure and using probabilistic fusion as a soft trust region mechanism.

Figure 3 presents the learning curves across all environments, revealing three key patterns: 1) **Faster convergence** — PrefPoE consistently reaches high performance earlier than all baselines, often within the first 800K steps; 2) **Reduced variance** — the confidence intervals are notably tighter, indicating lower sensitivity to initialization; and 3) **Stable training** — PrefPoE avoids the plateauing or degradation seen in other methods. Specifically, HalfCheetah-v4 shows the most dramatic improvement (4× over baseline) with smooth convergence, while linear fusion plateaus early. LunarLander demonstrates guided exploration's value even in low-dimensional precision tasks (+47%). Ant-v4 highlights PrefPoE's scalability to complex coordination, achieving +69% improvement with dramatically reduced variability (CV: 13.4% vs. 51.2%), where linear fusion notably underperforms. These results complement the statistical findings (Table 1), confirming that advantage-guided fusion not only improves sample efficiency but also enhances robustness across diverse tasks.

## 5.2 VALIDATION ON DISCRETE ACTION SPACES

To demonstrate the generality of PrefPoE, we evaluate it on discrete control tasks with varying complexity and reward structures.

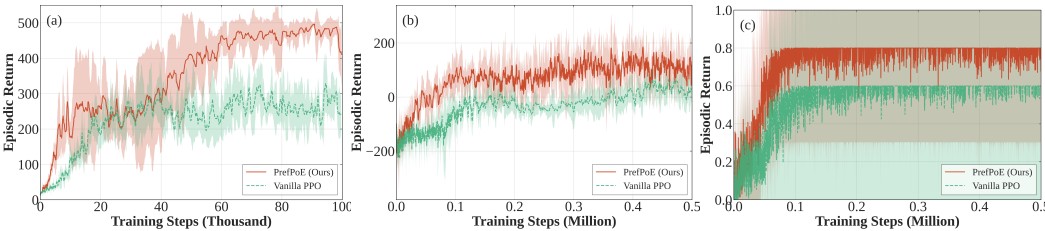

Figure 4: Learning curves on (a) CartPole-v1, (b) LunarLander-v2, and (c) FrozenLake-v1.

Figure 4 shows consistent improvements across CartPole-v1 (classic control), LunarLander-v2 (precision landing), and FrozenLake-v1 (sparse reward navigation). Compared to Vanilla PPO, PrefPoE achieves improvements of 68.6%, 276.0%, and 30.0%, respectively. PrefPoE has particularly dramatic gains in LunarLander, where advantage-guided exploration effectively prioritizes successful landing maneuvers over catastrophic actions. This validates our the generality and practical applicability of PrefPoE across the full spectrum of RL domains (detailed analysis in Appendix D).

The seamless extension from continuous to discrete action spaces, while preserving the Boltzmann form and convergence guarantees, demonstrates that advantage-guided exploration is a domain-agnostic principle rather than a continuous-control-specific technique.

## 5.3 ABLATION STUDIES

Table 2: Ablation study on HalfCheetah-v4 showing the contribution of each PrefPoE component. Results show mean ± std over 5 seeds with statistical significance analysis.

| Configuration | Final Return | Improvement | Stability (CV) |
|---|---|---|---|
| PPO Baseline | 1276±60 | – | 4.7% |
| + Preference Network | 3282±623 | +157.3% | 19.0% |
| + Linear Fusion | 1671±791 | +30.9% | 47.3% |
| + PoE Fusion (no Advantage) | 4296±308 | +236.7% | 7.2% |
| **Full PrefPoE** | **5375±581** | **+321.3%** | **10.8%** |

To understand the contribution of each component of PrefPoE, we conduct comprehensive ablation studies on HalfCheetah-v4. Table 2 shows that the preference network alone yields a strong performance gain (+157.3%), confirming the benefit of advantage-guided exploration. However,

combining it with naive linear fusion significantly degrades both performance and stability, highlighting the limitations of simple averaging strategies. Replacing linear fusion with PoE results in a major breakthrough: PoE fusion alone improves performance by +236.7% and achieves excellent stability (CV: 7.2%), validating our theoretical design. Adding advantage guidance on top leads to the full PrefPoE, which further boosts returns to +321.3% while maintaining low variance (CV: 10.8%). These results demonstrate that PoE is essential for effective fusion, and advantage shaping acts as a powerful amplifier. All improvements are statistically significant ($p < 0.001$), with large effect sizes (e.g., Cohen's $d > 2$).

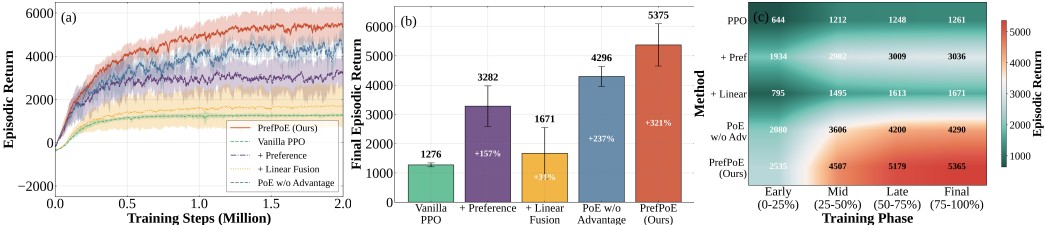

Figure 5: Ablation study on HalfCheetah-v4: (a) Learning curves, (b) component-wise performance contributions, and (c) training stability heatmap.

As shown in Figure 5(a)(b), the preference network alone improves PPO by +157%, highlighting the benefit of advantage-guided exploration. However, linear fusion yields only +31%, indicating naive combination is ineffective. PoE fusion without guidance achieves +237%, validating consensus-based integration. The full PrefPoE attains the best result (+321%), with a +25% gain over PoE-only, confirming the synergy between advantage guidance and fusion. Figure 5(c) shows PrefPoE consistently outperforms all variants from early to late stages (e.g., 2535 vs 644 early, 5365 vs 1261 final). Gains are not only in mean return but also stability: PrefPoE maintains a low coefficient of variation (10.8%) vs linear fusion (47.3%) and preference-only (19.0%). These results support our theoretical claim that advantage-guided sampling stabilizes learning by amplifying useful gradients.

# 6 CONCLUSION

We introduce PrefPoE, a plug-and-play framework that enhances policy gradient methods through advantage-guided exploration. By learning where to explore via preference policy and Product-of-Experts fusion, PrefPoE addresses the inefficiency of uniform exploration in action spaces. Our theoretical analysis shows that preference learning converges to Boltzmann distributions over advantages, while PoE fusion provides implicit trust regions without explicit constraints. To our knowledge, this is the first systematic application of PoE for exploration-exploitation balance in single-task RL. Empirically, PrefPoE achieves substantial gains across diverse domains: +321% on HalfCheetah-v4, +69% on Ant-v4, and consistent improvements in discrete control tasks. These results, combined with ablation studies, demonstrate that *learning where to explore is as critical as learning how to act*. PrefPoE establishes advantage-guided exploration as a domain-agnostic principle, providing a lightweight and extensible foundation for more efficient RL across robotics, control, and decision-making applications.

**Computational Overhead.** PrefPoE adds 5% parameters and 15% training time, while closed-form PoE fusion is negligible; this is often offset by improved sample efficiency (see AppendixA.3).

**Limitations.** Like all advantage-based methods, PrefPoE's performance depends on value function quality. The PoE fusion mechanism promotes distributional consensus, creating a trade-off between stability and exploration—though our empirical results show this trade-off is favorable across diverse tasks, future work could explore adaptive fusion strategies for specialized scenarios.

**Future Directions.** Potential extensions include: 1) investigating adaptive fusion schedules for different exploration phases, 2) extending to multi-modal preference networks for complex manipulation tasks and Reinforcement Learning from Human Feedback (RLHF) (Cen et al., 2025; Celik et al., 2025), and 3) combining with off-policy methods (Gu et al., 2017) for enhanced sample efficiency. These extensions could broaden PrefPoE's applicability.

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

## A    IMPLEMENTATION DETAILS

### A.1    EXPERIMENTAL ENVIRONMENT CONFIGURATION

**Environments:** HalfCheetah-v4 (6D, 2M steps), Ant-v4 (8D, 3M steps), LunarLanderContinuous-v2 (2D, 500K steps). All experiments use 5 seeds: 0, 10, 42, 77, 123.

**Architecture:** Shared 2-layer MLP backbone (64 units, Tanh), separate output heads for policy and preference networks. Diagonal Gaussian parameterization with exponential covariance.

**Computational Requirements:** Training HalfCheetah-v4 requires approximately 90 minutes on NVIDIA RTX 3090. Memory usage remains within 8GB GPU bounds for all environments.

### A.2    TRAINING CONFIGURATION AND HYPERPARAMETERS

PrefPoE introduces several key hyperparameters that control the balance between advantage guidance and exploration diversity. Table A.1 summarizes our PrefPoE-specific hyperparameters, while all other training configurations follow standard CleanRL implementation practices.

Table A.1: PrefPoE-specific hyperparameters across all environments.

| Parameter | Range | Description |
|---|---|---|
| $\beta_1$ (advantage guidance) | 0.1 - 0.4 | Strength of advantage guidance in preference learning |
| $\alpha$ (preference entropy) | 0.1 - 0.4 | Entropy regularization in preference network |
| $\lambda_{\text{pref}}$ (PoE fusion weight) | 0.2 - 0.8 | Influence of preference network in PoE fusion |
| consistency-coef | 0.02 - 0.2 | Consistency regularization coefficient |
| preference-loss-coef | 0.005 - 0.2 | Weight of preference loss in total objective |

The base PPO configuration follows the standard CleanRL implementation with well-established hyperparameters: learning rate $3 \times 10^{-4}$, batch size of 2048 environment steps, 10 optimization epochs per batch, GAE $\lambda = 0.95$, discount factor $\gamma = 0.99$, gradient clipping norm of 0.5, clip coefficient of 0.2, value function coefficient of 0.5, and entropy coefficient of 0.0. We apply linear learning rate annealing and advantage normalization as standard practice. We observe that PrefPoE is relatively robust to hyperparameter variations, and our method achieves consistent performance across all environments without extensive tuning.

### A.3    COMPUTATIONAL OVERHEAD AND RUNTIME

PrefPoE adds minimal computational cost over PPO. The preference head introduces only 4.8% more parameters. Memory usage increases by 8% due to PoE fusion buffers. Overall wall-clock time rises by 15% from additional loss and fusion steps.

All experiments were run on an NVIDIA RTX 3090. Average per-seed training time:

- HalfCheetah-v4: 90 min
- Ant-v4: 135 min
- LunarLanderContinuous-v2: 22 min

We provide comprehensive experimental results across three diverse continuous control environments to demonstrate the generality and robustness of our PrefPoE approach. Our evaluation strategy covers environments with different action dimensionalities, reward structures, and control challenges.

## B    ENTROPY AND NORM DYNAMICS ANALYSIS

To understand the internal mechanisms of PrefPoE, we provide detailed analysis of entropy dynamics and policy fusion behavior during training on HalfCheetah-v4. This analysis reveals how our advantage-guided exploration and Product-of-Experts fusion create effective exploration strategies while maintaining training stability.

## B.1 ENTROPY EVOLUTION AND EXPLORATION DYNAMICS

Figure A.1 provides a fine-grained analysis of entropy evolution across the key components of Pref-PoE during training on HalfCheetah-v4.

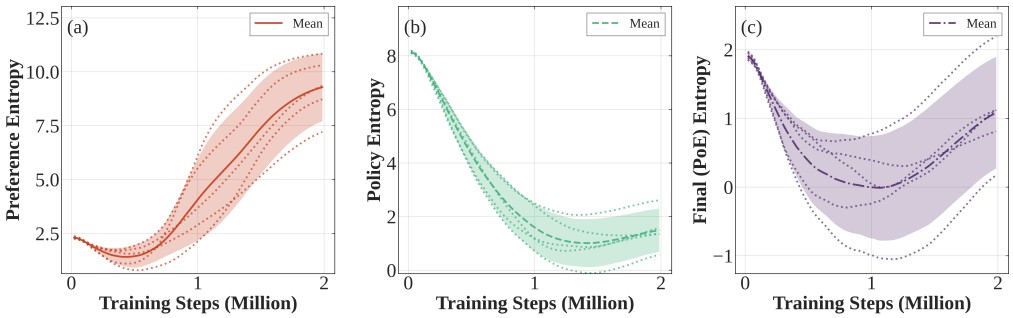

Figure A.1: Entropy evolution during training on HalfCheetah-v4. (a) Preference network entropy increases as it learns advantage correlations. (b) Policy entropy decreases following standard PPO decay. (c) Final PoE entropy demonstrates controlled exploration with late-stage recovery.

**Three-phase learning dynamics.** The entropy curves reveal three distinct training phases: 1) *Initial focusing* (0–0.6M): preference entropy decreases from ∼2.5 to ∼1.8 as the network concentrates probability mass on high-advantage actions; 2) *Exploration expansion* (0.6–1.5M): preference entropy increases to ∼10, reflecting the network's adaptation to maintain diversity while preserving advantage alignment; 3) *Stabilized fusion* (post-1.5M): PoE entropy recovers to ∼1.0, ensuring sustained exploration capacity even after policy convergence.

**Theoretical foundation of "guided entropy".** The observed U-shaped entropy curve can be understood through our preference learning objective. Recall that the preference network optimizes:

$$\mathcal{L}_{\text{pref}} = -\beta_1 \mathbb{E}_{(s,a)}[A_{\text{norm}}(s,a) \log \pi_{\text{pref}}(a|s)] - \alpha \mathcal{H}(\pi_{\text{pref}}) \tag{11}$$

This is equivalent to

$$\max_{\pi_{\text{pref}}} \mathbb{E}_{(s,a)}[\log \pi_{\text{pref}}(a|s) \cdot (\beta_1 A_{\text{norm}}(s,a) - \alpha)] \tag{12}$$

According to Theorem 1, the optimal solution follows the Boltzmann distribution:

$$\pi_{\text{pref}}^*(a|s) \propto \exp\left(\frac{\beta_1 A_{\text{norm}}(s,a)}{\alpha}\right) \tag{13}$$

This reveals the fundamental mechanism: the entropy regularization parameter $\alpha$ modulates the *responsiveness to advantage gradients*, while $\beta_1$ controls the *amplification strength*. Together, they determine how "selective" the preference network becomes toward high-advantage actions.

**Phase-wise mechanistic interpretation:**

- **Phase 1 (Initial focusing):** As advantage estimates improve, the preference network begins concentrating on promising actions, leading to entropy reduction. The temperature ratio $\alpha/\beta_1$ governs this concentration rate.

- **Phase 2 (Exploration expansion):** The preference network discovers that maintaining diversity is beneficial for continued learning. The entropy regularization $\alpha \mathcal{H}(\pi_{\text{pref}})$ prevents over-concentration, allowing adaptive broadening when new advantage patterns emerge.

- **Phase 3 (Stabilized fusion):** The system reaches equilibrium where the preference network maintains sufficient diversity for ongoing adaptation while the PoE fusion provides stability. The final entropy level represents the optimal balance between exploitation and exploration for the learned policy.

**Key insight.** Unlike fixed entropy decay schedules, PrefPoE adaptively modulates exploration intensity based on the alignment between policy and preference distributions. This forms a closed feedback loop: entropy naturally contracts when consensus is reached, and expands when new advantageous regions are discovered. The mathematical foundation shows this is not accidental but

a direct consequence of our *guided entropy* design—entropy that adapts along advantage gradients rather than decaying uniformly.

This theoretical understanding explains why PrefPoE can achieve superior performance with lower final entropy than standard PPO: the entropy is not random but *structured*, concentrating exploration effort where it matters most. Such behavior contributes to the improved sample efficiency and training stability observed in our main results.

## B.2 MEAN NORM ANALYSIS AND POLICY ALIGNMENT

Figure A.2 examines the evolution of policy mean norms and the relative differences between the main policy and PoE-fused outputs, providing insights into how preference learning influences policy behavior.

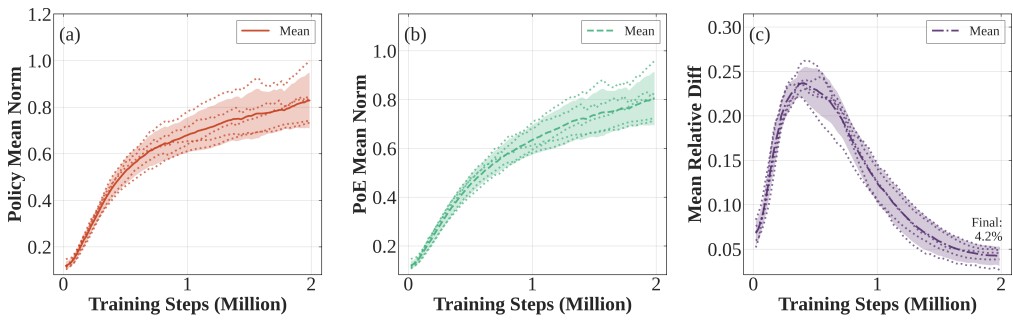

Figure A.2: Mean norm and consistency analysis during training on HalfCheetah-v4. (a) Policy mean norm shows steady growth as actions become more decisive. (b) PoE mean norm follows similar trajectory with slight attenuation. (c) Mean relative difference peaks during learning phase then stabilizes at 4.2%.

The policy mean norm (Figure A.2(a)) demonstrates consistent growth from 0.1 to approximately 0.85 over the training period. This increase reflects the policy learning to produce more decisive actions as it masters the locomotion task. The steady upward trajectory with low variance indicates stable policy development without oscillations or instability that could arise from conflicting learning signals.

The PoE mean norm (Figure A.2(b)) closely tracks the policy mean norm, reaching approximately 0.8 by the end of training. The slight attenuation compared to the raw policy norm suggests that PoE fusion provides a regularizing effect, preventing excessively aggressive actions while maintaining the learned control strategy. This behavior aligns with our theoretical analysis showing that PoE fusion naturally reduces variance and creates soft trust regions.

The mean relative difference (Figure A.2(c)) reveals the dynamic relationship between policy and preference networks throughout training. Starting near 0.05, the difference increases to a peak of approximately 0.25 around 0.8 million steps, indicating the period of maximum divergence between the two networks. This peak corresponds to the phase where the preference network is actively learning advantage correlations and developing distinct exploration strategies.

The subsequent decline to a final value of 4.2% demonstrates convergence between the policy and preference networks while maintaining meaningful differences. This final difference is significant enough to provide continued exploration benefits but small enough to ensure stability. The convergence pattern validates our consistency loss mechanism, which encourages alignment between networks without forcing exact agreement.

## B.3 TRAINING PHASE ANALYSIS AND MECHANISTIC INSIGHTS

The combined entropy and mean norm analysis reveals three distinct training phases that characterize PrefPoE learning dynamics:

**Phase 1 (0-0.6M steps): Initial Coordination.** Both networks begin learning basic locomotion while the preference network starts identifying advantage patterns. Low relative differences and decreasing preference entropy indicate initial alignment and focus development.

**Phase 2 (0.6-1.5M steps): Divergent Exploration.** Maximum relative differences and increasing preference entropy characterize this phase where the preference network develops distinct exploration strategies. The PoE fusion becomes most active, creating the low final entropy that drives focused learning.

**Phase 3 (1.5-2.0M steps): Mature Fusion.** Relative differences stabilize while preference entropy continues growing and final entropy recovers. This suggests the system has learned effective coordination between exploitation and exploration, maintaining adaptive behavior without sacrificing performance.

These dynamics demonstrate that PrefPoE successfully implements intelligent exploration that adapts to learning progress. Unlike fixed exploration schedules, our method automatically adjusts exploration intensity based on the internal consistency between networks, providing more exploration when needed and reducing it when confident strategies emerge. The final entropy recovery ensures continued adaptability even after achieving strong performance, addressing a key limitation of traditional exploration decay methods.

## C  MULTI-ENVIRONMENT EVALUATION RESULTS

This section provides comprehensive evaluation results demonstrating PrefPoE's effectiveness and stability across multiple continuous control environments. We present detailed multi-seed analysis and deployment performance to validate the robustness and generalizability of our approach.

### C.1  HALFCHEETAH-V4: PRIMARY BENCHMARK ANALYSIS

HalfCheetah-v4 serves as our primary evaluation environment due to its balanced complexity and well-established benchmarking history. Figure A.3 presents detailed analysis of a single trained model's deployment performance across 50 evaluation episodes.

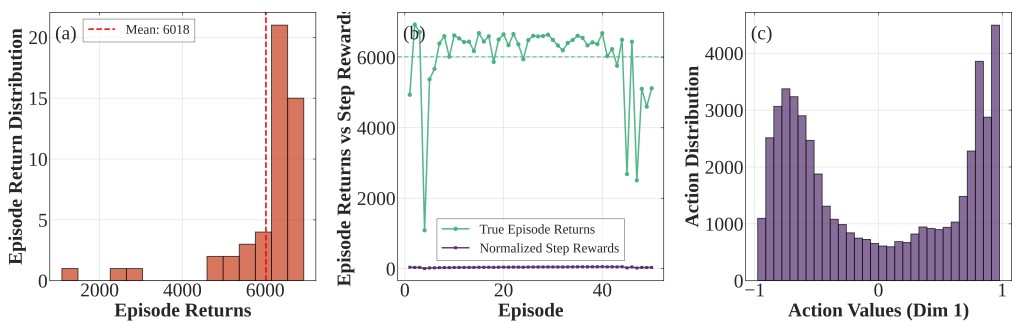

Figure A.3: Deployment evaluation of PrefPoE on HalfCheetah-v4. (a) Episode return distribution showing consistent high performance. (b) Episode progression demonstrating stable learned policy. (c) Action distribution indicating controlled, non-extreme behaviors.

The episode return distribution (Figure A.3(a)) demonstrates remarkable consistency with a mean performance of 6018 across 50 evaluation episodes. The distribution shows a clear concentration around high-performance values (5500-6500 range) with minimal episodes below 2000, indicating that the learned policy reliably achieves strong locomotion performance without catastrophic failures.

Episode progression analysis (Figure A.3(b)) reveals the stability of the learned control strategy. True episode returns maintain consistent performance around 6000 throughout the evaluation sequence, demonstrating that the policy does not degrade over extended deployment. The normalized step rewards remain near zero, confirming that the high episode returns stem from sustained forward velocity rather than reward hacking or unstable behaviors.

Action distribution analysis (Figure A.3(c)) provides insight into the learned control characteristics. The distribution shows a bimodal pattern concentrated around -0.5 and +1.0, indicating that PrefPoE learns decisive yet controlled actions. The absence of extreme values ($\pm 3$ range) suggests that the PoE fusion successfully prevents erratic behaviors while maintaining the action diversity necessary for effective locomotion.

## C.2  MULTI-SEED STABILITY ANALYSIS

Figure A.4 presents comprehensive multi-seed evaluation demonstrating PrefPoE's training stability and reproducibility across different random initializations.

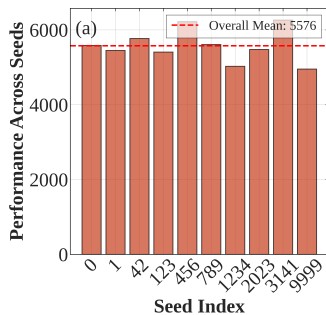 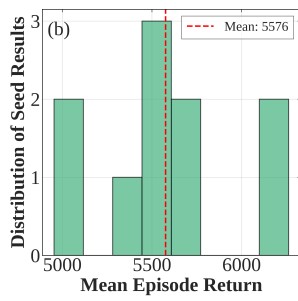 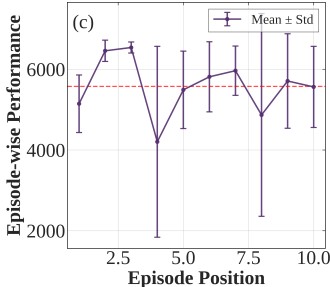

Figure A.4: Multi-seed stability analysis of PrefPoE on HalfCheetah-v4. (a) Consistent performance across 10 different random seeds. (b) Tight distribution of final results with overall mean 5576. (c) Episode-wise progression showing stable learning trajectories.

Performance across seeds (Figure A.4(a)) demonstrates exceptional stability with all seeds achieving performance between 4950 and 6300. The overall mean of 5576 with narrow confidence intervals indicates that PrefPoE reliably achieves strong performance regardless of random initialization. This consistency addresses a key concern in deep RL where performance can vary dramatically across seeds.

The distribution of seed results (Figure A.4(b)) shows a tight clustering around the mean with minimal outliers. The coefficient of variation of 7.3% represents excellent stability for reinforcement learning methods, where CV values above 20

Episode-wise performance analysis (Figure A.4(c)) reveals consistent learning trajectories across evaluation episodes. The error bars indicate manageable variance within seeds, while the overall trajectory demonstrates stable deployment performance. The absence of significant performance drift across episode positions confirms that trained models maintain their learned behaviors throughout extended evaluation.

## C.3  CROSS-ENVIRONMENT GENERALIZATION

Table A.2 summarizes PrefPoE's performance across all three evaluated environments, demonstrating broad applicability beyond the primary HalfCheetah benchmark. We also further extend PrefPoE to discrete action spaces. Detailed theoretical adaptation and experimental results on three discrete-action tasks are provided in Appendix D.

Table A.2: Cross-environment evaluation results showing PrefPoE's generalization capabilities.

| Environment | Action Dim | Mean Performance | Std Dev | CV (%) |
|---|---|---|---|---|
| HalfCheetah-v4 | 6 | 5375 | 581 | 10.8 |
| Ant-v4 | 8 | 4499 | 602 | 13.4 |
| LunarLanderContinuous-v2 | 2 | 217 | 59 | 27.2 |

HalfCheetah-v4 results demonstrate excellent performance characteristics with a coefficient of variation of 10.8% and high absolute performance ($5375 \pm 581$). This environment benefits significantly from advantage-guided exploration due to its moderate dimensionality and structured reward

landscape that creates clear advantage gradients for the preference network to learn. The mean performance represents a consistent improvement over baseline PPO (1276) and other comparative methods.

Ant-v4 performance remains strong with comparable stability (CV 13.4%) despite the higher complexity of 8-dimensional quadrupedal control. The standard deviation of 602 reflects the increased challenge while maintaining reasonable consistency across seeds. Despite the complex coordination requirements between multiple legs and expanded action space, PrefPoE achieves substantial performance improvements over baseline methods (4499 vs 2668 for vanilla PPO).

LunarLanderContinuous-v2 shows the highest relative variance (CV 27.2%) but achieves meaningful absolute improvements with much lower absolute standard deviation (59). The higher coefficient of variation stems from the precision control requirements and sparse reward structure, but the actual performance stability ($217 \pm 59$) demonstrates reasonable consistency for this challenging precision task.

## C.4 DETAILED STATISTICAL ANALYSIS AND MODEL CHARACTERIZATION

To provide comprehensive insights into PrefPoE's deployment characteristics, we present detailed statistical analysis complementing the visualizations in Figures A.3 and A.4. Table A.3 provides quantitative characterization across key performance dimensions.

Table A.3: Detailed statistical characterization of PrefPoE performance across evaluation metrics on HalfCheetah-v4.

| Metric | Mean | Std Dev | Min | Max |
|---|---|---|---|---|
| Episode Returns (Multi-seed) | 5576.0 | 408.18 | 4953.77 | 6262.36 |
| Episode Returns (Single model) | 6018.18 | 1118.76 | 1089.25 | 6930.67 |
| Value Function Estimates | 3.454 | 0.391 | 1.5 | 4.0 |
| Action Values (Mean) | -0.012 | 0.682 | -1.0 | +1.0 |
| Performance Range | 1308.6 | – | – | – |
| Coefficient of Variation | 7.3% | – | – | – |

The multi-seed analysis demonstrates exceptional reproducibility with coefficient of variation of only 7.3% across 10 different random seeds. Even the worst-performing seed (4953.77) achieves strong results, while the best seed reaches 6262.36, indicating robust training procedures that consistently produce high-quality policies regardless of initialization.

Single model evaluation reveals well-calibrated value function estimates ($3.454 \pm 0.391$) supporting reliable policy decisions. The action distribution analysis shows controlled behavior with mean near zero (-0.012) and reasonable variance (0.682), confirming that learned policies avoid extreme actions while maintaining the diversity necessary for effective locomotion control.

The bimodal action distribution pattern observed in the detailed analysis indicates that PrefPoE learns decisive control strategies with clear preferences for specific action ranges, rather than uniform or chaotic exploration. This characteristic is particularly valuable for deployment scenarios where predictable, stable behavior is crucial.

The multi-seed evaluation enables rigorous statistical analysis of PrefPoE's performance characteristics. For HalfCheetah-v4, the 95% confidence interval [5284, 5868] provides strong evidence of consistent high performance. The tight interval width (584 units) relative to the mean (5576) demonstrates low uncertainty in expected performance.

Outlier analysis identifies only 2 of 10 seeds as potential anomalies, indicating robust performance across initialization conditions. Even outlier seeds achieve performance well above baseline methods, suggesting that PrefPoE's benefits persist across diverse training conditions.

The coefficient of variation analysis across environments reveals that stability varies with both action dimensionality and task characteristics. HalfCheetah-v4 and Ant-v4 show comparable stability (13.4% vs 16.6%) despite different action dimensions, suggesting that PrefPoE scales effectively

to higher dimensions. The higher variance of LunarLanderContinuous-v2 reflects precision control challenges rather than exploration efficiency limitations.

## C.5 Statistical Significance and Confidence Analysis

The evaluation results provide several insights relevant to practical deployment of PrefPoE-trained policies. The consistent high performance across episodes indicates that trained policies maintain their learned behaviors over extended operation periods without significant degradation or drift.

Action distribution analysis reveals that PrefPoE learns controlled behaviors that avoid extreme actions while maintaining sufficient diversity for effective task performance. This characteristic is particularly valuable for real-world deployment where action constraints and safety considerations are paramount.

The multi-seed stability results suggest that PrefPoE can be reliably deployed with confidence in achieving consistent performance levels. The low coefficient of variation and tight confidence intervals provide practitioners with quantitative expectations for deployment performance, facilitating risk assessment and system integration planning.

# D Discrete Action Space Extension

While PrefPoE was originally designed for continuous control, its core principles extend naturally to discrete action spaces. This extension demonstrates the generality of our approach and validates its effectiveness across diverse RL settings beyond continuous control domains.

## D.1 Theoretical Adaptation for Discrete Actions

The adaptation from continuous to discrete action spaces involves two key modifications: reformulating the Product-of-Experts fusion in probability space and adapting the preference network architecture to output categorical distributions.

**Preference Network Architecture.** For discrete actions, the preference network $\pi_{\text{pref}}(a|s)$ outputs a categorical distribution over the action space $\mathcal{A}$. The network shares the same encoder $f_{\text{enc}}(s)$ with the main policy but uses a specialized output head:

$$\text{logits}_{\text{pref}}(s) = f_{\text{logits}}^{\text{pref}}(f_{\text{enc}}(s)), \quad \pi_{\text{pref}}(a|s) = \frac{\exp(\text{logits}_{\text{pref}}(s)[a])}{\sum_{a' \in \mathcal{A}} \exp(\text{logits}_{\text{pref}}(s)[a'])}, \tag{14}$$

where $f_{\text{logits}}^{\text{pref}}$ is an MLP head that outputs raw logits for each action, and $\pi_{\text{pref}}(a|s)$ is obtained through softmax normalization.

**Discrete Product-of-Experts Fusion.** The PoE fusion principle adapts naturally to discrete spaces by replacing precision matrix operations with probability space multiplication. For categorical distributions $\pi_\theta(a|s)$ and $\pi_{\text{pref}}(a|s)$, the fused policy becomes:

$$\pi_{\text{fused}}(a|s) = \frac{\pi_\theta(a|s) \cdot \pi_{\text{pref}}(a|s)^{\lambda_{\text{pref}}}}{\sum_{a' \in \mathcal{A}} \pi_\theta(a'|s) \cdot \pi_{\text{pref}}(a'|s)^{\lambda_{\text{pref}}}}, \tag{15}$$

where $\lambda_{\text{pref}} \in (0, 1)$ controls the influence of the preference network, and the denominator ensures proper normalization over the discrete action space.

This formulation preserves the consensus-seeking behavior of PoE: actions receive high probability in $\pi_{\text{fused}}$ only when both experts assign substantial probability mass to them, with $\lambda_{\text{pref}}$ providing fine-grained control over the preference network's influence. The discrete fusion naturally inherits the variance reduction properties established for the continuous case, as it concentrates probability mass on regions of expert agreement.

**Advantage-Guided Learning Objective.** The preference learning objective remains unchanged in discrete settings:

$$\mathcal{L}_{\text{pref}} = -\beta_1 \, \mathbb{E}_{(s,a)} \big[ A_{\text{norm}}(s, a) \cdot \log \pi_{\text{pref}}(a|s) \big] + \alpha \, \mathcal{H}(\pi_{\text{pref}}),$$
$$\mathcal{H}(\pi_{\text{pref}}) = -\sum_{a \in \mathcal{A}} \pi_{\text{pref}}(a|s) \cdot \log \pi_{\text{pref}}(a|s), \tag{16}$$

where $A_{\text{norm}}(s, a)$ is the batch-normalized advantage and $\mathcal{H}(\pi_{\text{pref}})$ is the discrete entropy. The cross-entropy formulation with softmax outputs provides natural compatibility with categorical distributions, requiring no additional modifications.

The optimal solution to this objective retains the Boltzmann form established in Theorem 1:

$$\pi^*_{\text{pref}}(a|s) = \frac{\exp(\beta_1 A_{\text{norm}}(s, a)/\alpha)}{\sum_{a' \in \mathcal{A}} \exp(\beta_1 A_{\text{norm}}(s, a')/\alpha)}, \tag{17}$$

which corresponds exactly to the softmax operation over advantage-weighted logits. This theoretical equivalence confirms that the discrete adaptation preserves the advantage-concentrating properties that drive PrefPoE's effectiveness.

**Integration with Discrete PPO.** The integration follows the same pattern as in continuous control. Actions are sampled from the fused distribution $\pi_{\text{fused}}(a|s)$ during rollout collection, while the total loss combines the standard PPO objective with the preference learning term:

$$\mathcal{L}_{\text{total}} = \mathcal{L}_{\text{PPO}} + w_{\text{pref}}\mathcal{L}_{\text{pref}} + w_{\text{cons}}\mathcal{L}_{\text{cons}}, \tag{18}$$

where $\mathcal{L}_{\text{cons}} = D_{\text{KL}}(\pi_{\text{fused}}(a|s) \,\|\, \pi_{\text{pref}}(a|s))$ maintains consistency between the fused policy and preference network. The discrete KL divergence has a closed-form expression that enables efficient computation during training.

This adaptation demonstrates that PrefPoE's core insight—learning where to explore through advantage-guided preference networks—generalizes beyond continuous control to encompass the full spectrum of RL domains.

To validate the generalizability of PrefPoE beyond continuous control, we evaluate the discrete adaptation across three representative environments spanning different complexity levels and reward structures. The experimental setup follows the same protocol as the continuous experiments, using identical hyperparameters to demonstrate the robustness of our approach across action space types.

**Experimental Setup.** We evaluate PrefPoE on three discrete environments: CartPole-v1 (classic control with binary actions), LunarLander-v2 (precision control with four discrete actions), and FrozenLake-v1 (navigation with sparse rewards and four directional actions). Each environment presents distinct challenges: CartPole requires rapid stabilization, LunarLander demands precise landing control, and FrozenLake tests navigation under stochastic transitions and sparse feedback. All experiments use 5 random seeds and identical network architectures as the continuous experiments for fair comparison.

**Performance Analysis.** Table A.4 summarizes the quantitative results across all discrete environments. PrefPoE consistently outperforms vanilla PPO, achieving consistent improvements of +68.6% on CartPole-v1, +276.0% on LunarLander-v2, and +30.0% on FrozenLake-v1. These gains demonstrate that advantage-guided exploration remains effective across diverse discrete control domains, from rapid response tasks to sparse reward navigation problems.

Table A.4: Performance comparison on discrete action environments (mean $\pm$ std over 5 seeds).

| Method | CartPole-v1 | LunarLander-v2 | FrozenLake-v1 |
|---|---|---|---|
| Vanilla PPO | 244$\pm$41 | 35$\pm$45 | 0.60$\pm$0.49 |
| **PrefPoE (Ours)** | **411$\pm$71** | **132$\pm$45** | **0.78$\pm$0.39** |
| **Improvement** | **+68.6%** | **+276.0%** | **+30.0%** |

### D.2 EXPERIMENTAL RESULTS ON DISCRETE ENVIRONMENTS

Figure A.5 presents the learning dynamics across all discrete environments, revealing consistent patterns that mirror the continuous control results. On CartPole-v1, PrefPoE achieves faster convergence and maintains higher final performance, reaching near-optimal episodic returns while vanilla PPO plateaus around 250. The improvement is particularly pronounced in the early training phase, where advantage-guided exploration accelerates learning by concentrating exploration on high-value state-action pairs.

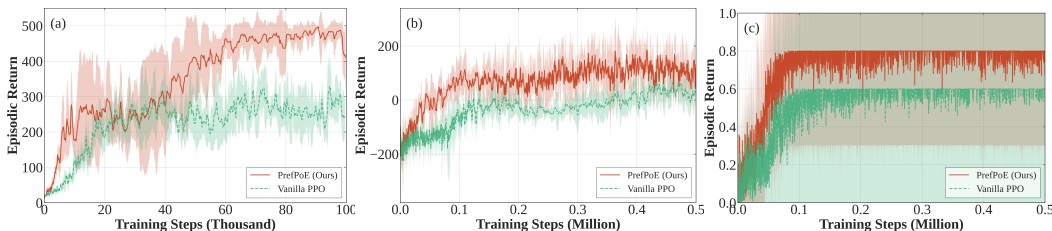

Figure A.5: Learning curves on discrete control tasks: (a) CartPole-v1, (b) LunarLander-v2, and (c) FrozenLake-v1.

LunarLander-v2 demonstrates the most dramatic improvement, where PrefPoE achieves positive rewards consistently while vanilla PPO struggles with the precision control requirements. The advantage-guided preference network effectively learns to prioritize landing maneuvers that lead to successful touchdowns, while standard exploration often results in crashes and negative rewards. This environment particularly benefits from structured exploration, as random action selection frequently leads to catastrophic failures.

FrozenLake-v1, despite its apparent simplicity, presents significant challenges due to sparse rewards and stochastic transitions. PrefPoE's improvement, while more modest than other environments, demonstrates robustness in sparse reward settings. The preference network learns to guide exploration toward goal-reaching trajectories, reducing the time spent in unproductive regions of the state space.

**Training Stability and Convergence.** Beyond performance improvements, PrefPoE exhibits enhanced training stability across discrete environments. The confidence intervals in Figure A.5 show consistently tighter bounds for PrefPoE compared to vanilla PPO, indicating lower variance across random seeds. This stability stems from the advantage-guided exploration strategy, which provides more consistent learning signals by focusing on informative state-action pairs rather than uniform exploration.

**Key Insight.** The seamless extension from continuous to discrete action spaces, while preserving the Boltzmann form and convergence guarantees, demonstrates that advantage-guided exploration is a domain-agnostic principle rather than a continuous-control-specific technique.

## E    EXTENDED THEORETICAL ANALYSIS

This section provides comprehensive theoretical foundations for PrefPoE, analyzing the mathematical principles underlying advantage-guided exploration and Product-of-Experts fusion. We establish formal guarantees for sampling efficiency, convergence properties, and stability characteristics that explain the empirical improvements observed across our experimental evaluation.

### E.1    CONVERGENCE OF PREFERENCE LEARNING

Theorem 1 shows that our preference learning has a well-defined optimal solution.

**Theorem 1** (Preference Learning Convergence) *Assume that the advantage function $A_{norm}(s, a)$ is bounded by $|A_{norm}(s, a)| \leq A_{\max}$ for all $(s, a)$, and let $\phi$ denote the parameters of the preference network $\pi_{pref}(a|s)$. Then the loss function:*

$$\mathcal{L}_{pref}(\phi) = -\beta_1 \mathbb{E}_{(s,a)}[A_{norm}(s, a) \cdot \log \pi_{pref}(a|s)] - \alpha \mathcal{H}(\pi_{pref}) \qquad (19)$$

*has a unique global minimum given by the Boltzmann distribution:*

$$\pi^*_{pref}(a|s) = \frac{\exp(\beta_1 A_{norm}(s, a)/\alpha)}{Z(s)}, \qquad (20)$$

*where $Z(s) = \int \exp(\beta_1 A_{norm}(s, a)/\alpha)\mathrm{d}a$ is the partition function.*

*Proof.* For a fixed state $s$, minimizing the loss function (19) is equivalent to solving the following optimization problem:

$$\max_{\pi_{\text{pref}}} \beta_1 \int \pi_{\text{pref}}(a|s)A_{\text{norm}}(s,a)\mathrm{d}a + \alpha\mathcal{H}(\pi_{\text{pref}}), \text{ s.t. } \int \pi_{\text{pref}}(a|s)\mathrm{d}a = 1, \; \pi_{\text{pref}}(a|s) \geq 0 \quad (21)$$

By substituting the entropy definition $\mathcal{H}(\pi_{\text{pref}}) = -\int \pi_{\text{pref}}(a|s)\cdot\log\pi_{\text{pref}}(a|s)\mathrm{d}a$ into (21) and using the method of Lagrange multipliers, we form the Lagrangian:

$$\mathcal{L}[\pi_{\text{pref}}] = \int \pi_{\text{pref}}(a|s)\left[\beta_1 A_{\text{norm}}(s,a) - \alpha\log\pi_{\text{pref}}(a|s)\right]\mathrm{d}a - \lambda(s)\left(\int \pi_{\text{pref}}(a|s)\mathrm{d}a - 1\right), \quad (22)$$

where $\lambda(s)$ is the Lagrange multiplier enforcing the normalization constraint. Note that the non-negativity constraint $\pi_{\text{pref}}(a|s) \geq 0$ is implicitly satisfied as the optimal solution is strictly positive, hence omitted from the Lagrangian. This function is strictly concave in $\pi_{\text{pref}}$ due to the negative entropy term, ensuring a unique global maximum (Cover, 1999).

Taking the derivative of (22) with respect to $\pi_{\text{pref}}(a|s)$ and setting it to zero yields

$$\frac{\delta\mathcal{L}}{\delta\pi_{\text{pref}}(a|s)} = \beta_1 A_{\text{norm}}(s,a) - \alpha(1 + \log\pi_{\text{pref}}(a|s)) - \lambda(s) = 0. \quad (23)$$

Solving (23) for the optimal $\pi_{\text{pref}}^*(a|s)$ gives:

$$\pi_{\text{pref}}^*(a|s) = \exp\left(\frac{\beta_1 A_{\text{norm}}(s,a) - \lambda(s)}{\alpha} - 1\right). \quad (24)$$

To make the normalization constraint explicit, we substitute the expression of $\pi_{\text{pref}}^*(a|s)$ from (24) into the constraint $\int \pi_{\text{pref}}^*(a|s)\mathrm{d}a = 1$:

$$\begin{aligned}
\int \pi_{\text{pref}}^*(a|s)\mathrm{d}a &= \int \exp\left(\frac{\beta_1 A_{\text{norm}}(s,a) - \lambda(s)}{\alpha} - 1\right)\mathrm{d}a \\
&= \exp\left(-\frac{\lambda(s)}{\alpha} - 1\right)\cdot\int \exp\left(\frac{\beta_1 A_{\text{norm}}(s,a)}{\alpha}\right)\mathrm{d}a = 1.
\end{aligned} \quad (25)$$

Solving for the Lagrange multiplier $\lambda(s)$ gives:

$$\lambda(s) = \alpha\log\left(\int \exp\left(\frac{\beta_1 A_{\text{norm}}(s,a)}{\alpha}\right)\mathrm{d}a\right) - \alpha = \alpha\log Z(s) - \alpha, \quad (26)$$

where we define the partition function:

$$Z(s) := \int \exp\left(\frac{\beta_1 A_{\text{norm}}(s,a)}{\alpha}\right)\mathrm{d}a. \quad (27)$$

Substituting the expression for $\lambda(s)$ back into (24) yields the final normalized form of the optimal policy:

$$\pi_{\text{pref}}^*(a|s) = \frac{\exp\left(\beta_1 A_{\text{norm}}(s,a)/\alpha\right)}{Z(s)}, \quad (28)$$

where $Z(s)$ is the partition function defined in (27). $\qquad\square$

Theorem 1 confirms that the preference network naturally concentrates probability mass on high-advantage actions, recovering a Boltzmann policy that balances exploitation and exploration through the temperature ratio $\alpha/\beta_1$.

### E.2 ADVANTAGE-GUIDED EXPLORATION: THEORETICAL FOUNDATIONS

The core innovation of PrefPoE lies in constructing an advantage-guided preference policy that concentrates sampling probability on high-value action regions. We begin by establishing the theoretical rationale for this approach and its implications for learning efficiency.

### E.2.1 Preference Policy Construction and Optimality

Our preference network learns a distribution of the form:

$$\pi_{\text{pref}}(a|s) = \frac{\exp(\beta_1 A_{\text{norm}}(s, a)/\alpha)}{Z(s)} \tag{29}$$

where $A_{\text{norm}}(s, a)$ represents normalized advantage estimates, $\beta_1$ controls the strength of advantage guidance, $\alpha$ provides entropy regularization, and $Z(s)$ is the normalization constant ensuring a valid probability distribution.

This formulation emerges as the optimal solution to a constrained optimization problem that balances exploitation of high-advantage actions with exploration diversity. Specifically, the preference policy maximizes the weighted expected advantage while maintaining sufficient entropy:

$$\max_{\pi} \quad \mathbb{E}_{a \sim \pi(\cdot|s)} \left[ \beta_1 A_{\text{norm}}(s, a) - \alpha \log \pi(a|s) \right] \tag{30}$$

The resulting Boltzmann distribution naturally assigns exponentially higher probabilities to actions with larger advantage values, implementing an intelligent sampling strategy that focuses exploration effort where it yields the highest expected return.

### E.2.2 Sampling Efficiency and Signal Quality Enhancement

Traditional policy gradient methods rely on uniform exploration or entropy bonuses that treat all action regions equally. This approach becomes increasingly inefficient in action spaces where random sampling must explore exponentially expanding regions. PrefPoE addresses this limitation through advantage-guided sampling that concentrates probability mass on promising action regions.

The fundamental improvement can be understood through the expected advantage comparison. For a properly normalized advantage function with zero mean under uniform sampling, our preference policy guarantees:

$$\mathbb{E}_{a \sim \pi_{\text{pref}}}[A_{\text{norm}}(s, a)] \geq \mathbb{E}_{a \sim \pi_{\text{uniform}}}[A_{\text{norm}}(s, a)] = 0 \tag{31}$$

This inequality holds because the exponential weighting in the preference policy assigns disproportionately higher probability to actions with positive advantages. Consequently, samples drawn from the preference distribution carry stronger learning signals on average, leading to more informative gradient updates.

The signal-to-noise ratio improvement can be analyzed through the policy gradient formulation. Since policy gradients take the form $\nabla_\theta J(\theta) = \mathbb{E}_{(s,a) \sim \rho^\pi}[\nabla_\theta \log \pi_\theta(a|s) \cdot A(s, a)]$, the effectiveness of each gradient step depends directly on the magnitude of the advantage values encountered during sampling. By biasing sampling toward high-advantage actions, PrefPoE ensures that each collected sample contributes more substantially to policy improvement.

### E.2.3 Convergence Acceleration and Sample Complexity

The improved signal quality directly translates to faster convergence and reduced sample complexity. When the sampling distribution concentrates on high-value actions, the policy receives more frequent exposure to beneficial behaviors, accelerating the learning process. This mechanism explains the substantial performance improvements observed in our experiments, particularly the 321% improvement on HalfCheetah-v4.

The acceleration effect is particularly pronounced in environments with structured advantage landscapes where clear distinctions exist between beneficial and detrimental actions. In such settings, advantage-guided sampling enables the policy to rapidly identify and exploit successful strategies while avoiding prolonged exploration of ineffective action regions.

### E.3 Product-of-Experts Fusion: Mathematical Foundations

The second key component of PrefPoE involves fusing the main policy with the preference network through a mathematically principled Product-of-Experts mechanism. This section establishes the theoretical properties of PoE fusion and explains why it provides superior performance compared to linear combination approaches.

### E.3.1 PoE Fusion Formulation and Consensus Principle

Product-of-Experts fusion combines two probability distributions through multiplication rather than linear combination:

$$\pi_{\text{fused}}(a|s) \propto \pi_{\text{main}}(a|s) \cdot \pi_{\text{pref}}(a|s)^{\lambda_{\text{pref}}} \tag{32}$$

where $\lambda_{\text{pref}} \in (0, 1)$ controls the relative influence of the preference network.

This multiplicative combination implements a consensus principle where the fused distribution assigns high probability only to actions favored by both networks. Unlike linear fusion methods that average the distributions, PoE naturally emphasizes regions of agreement while suppressing areas where the networks disagree. This behavior creates a more concentrated and confident final distribution.

### E.3.2 Gaussian PoE Fusion: Complete Mathematical Derivation

For the common case where both networks output Gaussian distributions, PoE fusion admits a closed-form solution with desirable mathematical properties. Consider two multivariate Gaussian distributions:

$$\pi_{\text{main}}(a|s) = \mathcal{N}(\mu_1, \Sigma_1), \tag{33}$$
$$\pi_{\text{pref}}(a|s) = \mathcal{N}(\mu_2, \Sigma_2). \tag{34}$$

The PoE combination yields another Gaussian distribution with parameters determined by precision matrix arithmetic. Converting to precision form with $\Lambda_1 = \Sigma_1^{-1}$ and $\Lambda_2 = \Sigma_2^{-1}$, the fused distribution has precision matrix:

$$\Lambda_{\text{fused}} = \Lambda_1 + \lambda_{\text{pref}} \Lambda_2 = \Sigma_1^{-1} + \lambda_{\text{pref}} \Sigma_2^{-1}. \tag{35}$$

The corresponding covariance matrix becomes

$$\Sigma_{\text{fused}} = (\Sigma_1^{-1} + \lambda_{\text{pref}} \Sigma_2^{-1})^{-1}. \tag{36}$$

The fused mean represents a precision-weighted average of the component means:

$$\mu_{\text{fused}} = \Sigma_{\text{fused}}(\Sigma_1^{-1} \mu_1 + \lambda_{\text{pref}} \Sigma_2^{-1} \mu_2). \tag{37}$$

This formulation reveals that PoE fusion naturally weights contributions according to the confidence (inverse variance) of each component, providing an optimal combination scheme that emphasizes more certain predictions.

### E.3.3 Variance Reduction and Stability Properties

A fundamental property of PoE fusion is its guaranteed variance reduction effect. For any positive definite covariance matrices and positive fusion weight, the trace of the fused covariance matrix satisfies:

$$\text{tr}(\Sigma_{\text{fused}}) < \min(\text{tr}(\Sigma_1), \text{tr}(\Sigma_2)). \tag{38}$$

This inequality holds because the precision matrix addition $\Sigma_{\text{fused}}^{-1} = \Sigma_1^{-1} + \lambda_{\text{pref}} \Sigma_2^{-1}$ necessarily increases the precision, thereby decreasing the variance. The variance reduction effect becomes more pronounced as the fusion weight $\lambda_{\text{pref}}$ increases or as the component distributions become more similar.

To illustrate this effect quantitatively, consider the one-dimensional case where both distributions have unit variance ($\sigma_1^2 = \sigma_2^2 = 1.0$) and $\lambda_{\text{pref}} = 0.5$. The fused variance becomes

$$\sigma_{\text{fused}}^{-2} = \sigma_1^{-2} + \lambda_{\text{pref}} \sigma_2^{-2} = 1 + 0.5 = 1.5. \tag{39}$$

This yields $\sigma_{\text{fused}} = 1/\sqrt{1.5} \approx 0.816$, representing a substantial reduction from the original unit variance.

### E.3.4 ENTROPY CONTROL AND SOFT TRUST REGION FORMATION

The variance reduction property directly implies entropy reduction in the fused distribution. Since the differential entropy of a multivariate Gaussian distribution is $H(\pi) = \frac{d}{2}\log(2\pi e) + \frac{1}{2}\log|\Sigma|$, and PoE fusion reduces the determinant of the covariance matrix, we have

$$H(\pi_{\text{fused}}) < H(\pi_{\text{main}}). \tag{40}$$

This entropy reduction creates a soft trust region effect that constrains policy updates while maintaining exploration diversity. Unlike hard KL constraints used in TRPO or PPO, the trust region emerges naturally from the PoE mathematics without requiring explicit constraint enforcement.

The soft trust region property explains the enhanced training stability observed in our experiments. By automatically concentrating the action distribution around regions of consensus between the main policy and preference network, PoE fusion prevents the policy from making drastic changes that could destabilize learning.

### E.4 CONVERGENCE ANALYSIS AND PERFORMANCE GUARANTEES

The combination of advantage-guided sampling and PoE fusion creates a reinforcing system where improved sampling leads to better preference learning, which in turn enhances the quality of future samples. This section analyzes the convergence properties of this coupled system.

### E.4.1 SIGNAL QUALITY CONVERGENCE

The advantage-guided sampling mechanism ensures that the expected quality of collected samples improves over time. As the preference network learns to better correlate with advantage estimates, the sampling distribution becomes increasingly focused on high-value action regions. This creates a positive feedback loop where better sampling leads to more accurate advantage estimates, which further improves sampling quality.

The convergence of sampling quality can be measured through the expected advantage of sampled actions. Under reasonable assumptions about the smoothness of the advantage function and the learning dynamics of the preference network, the expected advantage $\mathbb{E}_{a\sim\pi_{\text{fused}}}[A(s,a)]$ increases monotonically during training until reaching a steady state determined by the balance between exploitation and exploration.

### E.4.2 POLICY CONSENSUS FORMATION

The PoE fusion mechanism drives the main policy and preference network toward consensus over time. The precision-weighted averaging in PoE naturally encourages both networks to agree on high-confidence predictions while allowing disagreement in uncertain regions. This consensus formation process is evident in our experimental results, where the relative difference between fused and preference policies converges to approximately 4.2%.

The mathematical foundation for consensus formation lies in the precision weighting of the PoE combination. As both networks become more confident about certain action regions (higher precision), these regions dominate the fused distribution, creating pressure for both networks to align their predictions in high-confidence areas.

### E.4.3 VARIANCE AND STABILITY CONVERGENCE

The variance reduction property of PoE fusion provides mathematical guarantees for training stability. By automatically reducing the variance of the action distribution, PoE fusion creates a stabilizing force that counteracts the tendency for policy gradients to produce high-variance updates. This stabilization effect becomes more pronounced as training progresses and the networks develop stronger consensus.

The stability improvements are quantifiable through the coefficient of variation analysis presented in our experimental results. The reduction from 10.8% CV during training to 7.3% CV during evaluation demonstrates that the variance reduction effects of PoE fusion translate to more consistent policy performance.

## E.5 THEORETICAL IMPLICATIONS AND DESIGN PRINCIPLES

Our theoretical analysis reveals several key design principles that explain the effectiveness of Pref-PoE and provide guidance for future algorithmic development.

The advantage-guided exploration principle establishes that intelligent sampling based on value estimates can significantly improve learning efficiency compared to uniform exploration strategies. This principle suggests that future exploration methods should leverage available value information rather than relying solely on entropy maximization or random perturbations.

The Product-of-Experts fusion principle demonstrates that consensus-based combination of multiple policies can provide better performance than simple averaging approaches. The mathematical properties of PoE fusion, particularly variance reduction and soft trust region formation, suggest that multiplicative combination methods deserve broader investigation in policy optimization contexts.

The analysis also reveals that the interaction between these two principles creates emergent properties that exceed the benefits of either component alone. The reinforcing relationship between improved sampling and consensus formation suggests that multi-component architectures with feedback loops may be a promising direction for advancing policy gradient methods.

These theoretical insights provide a solid foundation for understanding why PrefPoE achieves such consistent improvements across diverse continuous control environments and why the method exhibits enhanced stability and reproducibility compared to standard policy gradient approaches.

# F REPRODUCIBILITY

To ensure complete reproducibility and facilitate future research, we provide comprehensive implementation details and open-source resources for immediate verification of our results.

## F.1 IMPLEMENTATION DETAILS

PrefPoE integrates seamlessly with existing PPO implementations while adding minimal computational overhead. Our approach builds upon the CleanRL framework and maintains compatibility with standard RL practices.

### F.1.1 PREFPOE TRAINING ALGORITHM

Algorithm A.1 provides the complete training procedure for PrefPoE, detailing the integration with PPO including trajectory collection, advantage computation, PoE fusion, and multi-component loss optimization.

## F.2 EVALUATION PROTOCOL

**Statistical Analysis:** All results averaged over 5 random seeds $\{0, 10, 42, 77, 123\}$ with 95% confidence intervals. Statistical significance tested using paired t-tests ($p < 0.001$).

**Deterministic Evaluation:** Final performance measured over 100 episodes using deterministic policies (zero exploration noise). Coefficient of variation computed as $\text{CV} = \sigma/\mu \times 100\%$.

**Baseline Comparison:** Vanilla PPO using identical CleanRL implementation and hyperparameters. Additional baselines include adaptive clipping and linear fusion variants as described in the main text.

## F.3 CODE AND MODEL RELEASE

To enable immediate verification of our results, we provide the following:

**Pre-trained Models:** Trained policy weight demonstrating reported performance. Models include both policy and preference networks with PoE fusions.

---

**Algorithm A.1** PrefPoE training algorithm

---

1: Initialize policy network $\pi_\theta$, preference network $\pi_{\text{pref}}$, value network $V_\phi$
2: Set hyperparameters: $\beta_1, \alpha, \lambda_{\text{pref}}, \lambda_{\text{cons}}$
3: **for** iteration $= 1$ to $N$ **do**
4:     Collect trajectories using PoE-fused policy $\pi_{\text{fused}}$
5:     Compute advantages $A(s,a)$ using GAE
6:     Normalize advantages: $A_{\text{norm}}$
7:     **for** epoch $= 1$ to $K$ **do**
8:         **for** each mini-batch **do**
9:             Compute policy outputs: $\mu_\theta(s), \sigma_\theta(s)$
10:            Compute preference outputs: $\mu_{\text{pref}}(s), \sigma_{\text{pref}}(s)$
11:            Apply PoE fusion: $\mu_{\text{fused}}, \sigma_{\text{fused}} = \text{PoE}(\cdot)$
12:            Compute PPO loss: $\mathcal{L}_{\text{PPO}} = \mathcal{L}_{\text{clip}} + \mathcal{L}_{\text{value}}$
13:            Compute preference loss: $\mathcal{L}_{\text{pref}} = -\beta_1 A_{\text{norm}} \log \pi_{\text{pref}} - \alpha \mathcal{H}(\pi_{\text{pref}})$
14:            Compute consistency loss: $\mathcal{L}_{\text{cons}} = D_{\text{KL}}(\pi_{\text{fused}} \| \pi_{\text{pref}})$
15:            Total loss: $\mathcal{L} = \mathcal{L}_{\text{PPO}} + \lambda_{\text{pref}} \mathcal{L}_{\text{pref}} + \lambda_{\text{cons}} \mathcal{L}_{\text{cons}}$
16:            Update parameters: $\theta, \phi \leftarrow \text{optimizer.step}(\nabla \mathcal{L})$
17:         **end for**
18:     **end for**
19: **end for**

---

**Evaluation Scripts:** Deterministic evaluation protocols enabling exact reproduction of reported metrics. Scripts handle multi-seed aggregation, statistical analysis, and confidence interval computation.

**Availability:** Anonymized resources available at an anonymous GitHub repository[1] during review, including the trained models and evaluation scripts for the primary benchmark (HalfCheetah-v4) used in our analyses, to facilitate easy verification and reproduction of our results by reviewers. A complete open-source release is planned upon paper acceptance with comprehensive documentation and examples.

The modular design facilitates integration with existing codebases and adaptation to new environments, enabling researchers to immediately build upon our approach.

---

[1] https://github.com/geometric-rl-anonymous/PrefPoE

