# OpenReview forum: "PrefPoE: Advantage-Guided Preference Fusion for Learning Where to Explore"
_ICLR.cc/2026/Conference — ICLR 2026 Conference Withdrawn Submission_

### Official Review · Reviewer_FUjx · 2025-10-19

**Soundness:** 2
**Presentation:** 2
**Contribution:** 1
**Rating:** 2
**Confidence:** 3

**Summary:**

This paper proposes a method to improve sample efficiency in policy-based reinforcement learning by modifying the exploration strategy. The authors argue that exploitative exploration (favoring actions with higher advantages) outperforms entropy-based random exploration. Their technical approach involves training a separate preference head that weights actions according to their advantage estimates. During trajectory rollouts, the policy combines outputs from both the preference head and the standard policy head. Experimental results on six tasks demonstrate improvements over PPO-based methods with high-entropy exploration.

**Strengths:**

1. **Sound technical approach**: The use of PoF mechanism to reshape action distributions is technically sound. Incorporating an auxiliary preference head that considers advantage estimates provides an interpretable way to bias action selection toward potentially beneficial states.

**Weaknesses:**

1. **Weak Theoretical Motivation** The central claim that *exploration should be guided by advantage estimates rather than being uniformly distributed* lacks rigorous justification. This weakens the paper's foundational motivation. Moreover, the characterization of standard trajectory rollout as "uniformly sampling" is inaccurate that standard policy gradient methods already bias sampling toward actions with higher probabilities under the learned policy. Value information is implicitly incorporated through policy updates.
2. **Insufficient Support for Strong Claims** The paper makes claims that directly contradict established exploration literature (e.g., the benefits of entropy regularization for exploration). Such significant departures from accepted principles require: i) deeper theoretical analysis explaining when and why advantage-guided exploration outperforms entropy-based approaches; ii) clear statement of assumptions under which this holds; iii) justification for why existing mechanisms (e.g., tuning entropy coefficients, including negative values for more exploitative behavior) cannot address the identified issues.
3. **Limited Experimental Scope** The experimental evaluation only compares against PPO variants, which significantly limits its contribution. To substantiate the claims, the paper should include: i) modern policy gradient methods (e.g., TRPO, SAC, A3C) ii)  methods that explicitly balance exploration-exploitation (e.g., UCB-based approaches)

**Questions:**

1. Have you conducted a Pareto frontier analysis examining the exploration-exploitation trade-off? Plotting performance metrics against exploration behavior would provide more nuanced insights into when and why your method outperforms baselines.
2. Can you explain the specific failure mode of PPO that prevents it from exploring high-advantage actions? If PPO's policy updates are guided by advantage estimates (via its objective), why doesn't it naturally converge to such actions? What specific aspect of π_pref addresses this failure that standard policy optimization cannot?
3. How does your method compare to simply adjusting the entropy regularization coefficient (potentially to negative values for exploitation)? This seems like a more direct baseline for your approach.

---

### Official Review · Reviewer_Pgag · 2025-10-26

**Soundness:** 3
**Presentation:** 3
**Contribution:** 2
**Rating:** 2
**Confidence:** 4

**Summary:**

The authors introduce an advantage-guided exploration method using a product-of-experts fusion to balance exploration and exploitation. It is shown that the resulting algorithm outperforms PPO baselines by a substantial fraction. The authors provide nice intuitions about the workings of the EE balance, and some theoretical understanding is provided regarding the soft-max nature of the preferred policy.

**Strengths:**

The authors provide some theoretical understanding regarding the nature of the preference policy, that focuses exploration on the regions that are more promising according to the current estimates of the advantage function. Nice intuitions about how the exploration-exploitation works are provided.

**Weaknesses:**

Although a comparison with PPO is provided, a comparison with the state-of-the-art SAC is not provided. For instance, in Table 1-2 in reference

https://openreview.net/forum?id=HhbHw2yInZ

the authors can see that the SAC performance is larger than the one provided in the current article. Also, apparently, the PPO result for the ant is better than the one provided by the authors. I understand that there could be some implementation differences, but then it is unclear how robust results are to these choices and whether conclusions are generally valid or not in the current paper.

A comparison with the SAC baseline is critical because entropy-regularization in SAC also provides an exploration-exploitation balance that persists over the learning and does not collapse.

Further, the theoretical part of the paper is not very strong. The fact that advantage-guided policies are soft-max is a direct consequence of entropy regularization, already shown in

https://proceedings.mlr.press/v80/haarnoja18b

Indeed, one can see that advantage-guided policy is equivalent to a Q-guided policy, which is simply a reward function regularized by a policy entropy term, with some differences in normalization.

Further, SAC already has an EE tradeoff in the form of a SoftMax policy that favors actions with larger Q.

**Questions:**

See weaknesses.

---

### Official Review · Reviewer_JXbX · 2025-10-27

**Soundness:** 2
**Presentation:** 2
**Contribution:** 2
**Rating:** 2
**Confidence:** 4

**Summary:**

The paper presents an RL algorithm called PrefPoE based on PPO that samples action from a product of normal distribution of normal PPO and a Boltzmann distribution constructed with advantages.

**Strengths:**

1. The performance of PrefPoE is very strong compared to vannila PPO on the environments of choice.
2. The motivation is very clear

**Weaknesses:**

1. The motivation and background of the problem is clearly stated, however, the related work part does include the papers that are directly related to this problem:
* **Decoupled Exploration and Exploitation Policies for Sample-Efficient Reinforcement Learning (2021), by Whitney et al.**: This paper is most directly related to this PrefPoE, which uses the same "fused distribution" to sample actions. The only difference is that  they use a curiosity-based exploration policy for exploration instead of the entropy regularized exploration policy used by PrefPoE.
* **Hyper: Hyperparameter Robust Efficient Exploration in Reinforcement Learning (2025), by Wang et al.**: This paper shares the same motivation but uses two phases rollout to explore the high-reward region instead of sampling from a fused distribution.
* **Decoupling Exploration and Exploitation in Reinforcement Learning (2021), by Schäfer et al.**
The authors should discuss the papers mentioned above, and there may be other papers also related on preventing the over-exploration.

2. The shown results are strong, but clearly **insufficient to conclude that PrefPoE is a good exploration strategy**. The only conclusion I managed to conclude is the following: **PrefPoE is better than vanilla PPO on dense reward environment**.
The choice of environments in this work is biased: HalfCheetah and Ant are two environment with most dense reward in MuJoCo domain, as there are less absorbing / terminal state compared to Hopper, Walker and Humanoid. And the experiments on Cartpole, LunarLander and FrozenLake are even simpler.

**Questions:**

1. Whether the algo can perform well on sparse-reward environments? As the authors claimed PrefPoE is a good **exploration** strategy, it is necessary to prove it can explore in hard problems. Including the missing domains on MuJoCo (Hopper, Walker, Humanoid), as well as the sparse version of MuJoCo.

2. Is the technique compatible with off-policy algorithms with guassian policy head like SAC?

---

### Official Review · Reviewer_5UZN · 2025-10-30

**Soundness:** 2
**Presentation:** 3
**Contribution:** 2
**Rating:** 2
**Confidence:** 4

**Summary:**

This paper proposes PrefPoE, a reinforcement learning framework that enhances exploration in policy gradient methods. PrefPoE introduces a Preference-Product-of-Experts (PoE) mechanism that fuses a main policy with a preference policy trained to focus probability mass on high-advantage actions. This results in more targeted exploration, aiming to balance exploration and exploitation more efficiently. The authors provide theoretical analysis showing that the preference policy converges to a Boltzmann distribution over advantages and that PoE fusion induces a “soft trust region.” Empirical results on standard continuous control tasks (HalfCheetah, Ant, LunarLander) and a few discrete tasks demonstrate substantial performance improvements over vanilla PPO.

**Strengths:**

1. Exploration remains a central challenge in RL, and the idea of guiding exploration based on advantage estimates is both intuitive and meaningful.

2. The paper provides solid algorithmic detail and theoretical justification, making the contribution more credible.

3. Experiments across multiple environments show strong performance gains and stability improvements.

4. The authors analyze the contribution of core components, which helps clarify the source of performance improvement.

**Weaknesses:**

1. The tested environments are mostly standard locomotion tasks (HalfCheetah, Ant, LunarLander) with relatively simple reward landscapes. PrefPoE may perform well there, but it remains unclear how it handles more complex or multimodal tasks (e.g., Humanoid or HumanoidBench tasks [1]) where local optima and sparse rewards dominate.

2. The paper focuses mainly on PPO-based variants. Other exploration-focused methods (e.g., RND [2], ICM [3], or parameter noise approaches) are not compared. This makes it hard to judge whether the proposed method truly advances exploration research rather than just improving PPO training dynamics.

3. PrefPoE introduces several new hyperparameters (e.g., β₁, α, λ_pref, w_pref, w_cons). The paper claims robustness but provides no systematic sensitivity analysis, which limits its practical usability.

4. The paper suggests domain-agnostic applicability (to discrete and continuous actions), but the discrete experiments are minimal and shallow, leaving this claim only partially supported.

[1] Sferrazza, Carmelo; Huang, Dun-Ming; Lin, Xingyu; Lee, Youngwoon; Abbeel, Pieter. HumanoidBench: Simulated Humanoid Benchmark for Whole-Body Locomotion and Manipulation. arXiv preprint arXiv:2403.10506, 2024.

[2] Random Network Distillation (RND):
Burda, Yuri; Edwards, Harrison; Storkey, Amos; Klimov, Oleg. Exploration by Random Network Distillation. In 7th International Conference on Learning Representations (ICLR 2019).

[3] Intrinsic Curiosity Module (ICM):
Pathak, Deepak; Agrawal, Pulkit; Efros, Alexei A.; Darrell, Trevor. Curiosity-driven Exploration by Self-supervised Prediction. In International Conference on Machine Learning (ICML 2017).

**Questions:**

1. Have you compared PrefPoE with PPO using tuned entropy coefficients or adaptive entropy schedules to ensure that improvements are not merely due to different exploration temperatures?

2. Can you provide experiments on higher-dimensional and more complex tasks (e.g., Humanoid, HumanoidBench)?

3. How sensitive is performance to the new hyperparameters introduced?

4. How does PrefPoE compare with novelty-based exploration methods (e.g., RND, ICM) or parameter-space noise approaches?

5. (Minor) Please correct the citation for vanilla PPO (line 342) — it should refer to Schulman et al., “Proximal Policy Optimization Algorithms,” 2017.

---

### Note · Authors · 2025-11-13

I have read and agree with the venue's withdrawal policy on behalf of myself and my co-authors.